# Antibiotic Use, Incidence and Risk Factors for Orthopedic Surgical Site Infections in a Teaching Hospital in Madhya Pradesh, India

**DOI:** 10.3390/antibiotics11060748

**Published:** 2022-05-31

**Authors:** Kristina Skender, Anna Machowska, Vivek Singh, Varun Goel, Yogyata Marothi, Cecilia Stålsby Lundborg, Megha Sharma

**Affiliations:** 1Department of Global Public Health, Health Systems and Policy, Karolinska Institutet, 17177 Stockholm, Sweden; anna.machowska@ki.se (A.M.); cecilia.stalsby.lundborg@ki.se (C.S.L.); megha.sharma@ki.se (M.S.); 2Department of Orthopedics, Ruxmaniben Deepchand Gardi Medical College, Surasa, Ujjain 456006, India; drviveksingh29@rediffmail.com (V.S.); cool.varungoel@yahoo.com (V.G.); 3Department of Microbiology, Ruxmaniben Deepchand Gardi Medical College, Surasa, Ujjain 456006, India; dryogyata@yahoo.com; 4Department of Pharmacology, Ruxmaniben Deepchand Gardi Medical College, Surasa, Ujjain 456006, India

**Keywords:** surgical site infections, SSI, incidence, risk factors, orthopedic, antibiotic susceptibility patterns, private, teaching, tertiary care hospital, India

## Abstract

Orthopedic surgeries contribute to the overall surgical site infection (SSI) events worldwide. In India, SSI rates vary considerably (1.6–38%); however, there is a lack of a national SSI surveillance system. This study aims to identify the SSI incidence, risk factors, antibiotic prescription and susceptibility patterns among operated orthopedic patients in a teaching hospital in India. Data for 1205 patients were collected from 2013 to 2016. SSIs were identified based on the European Centre for Disease Prevention and Control guidelines. The American Society for Anesthesiologists classification system was used to predict patients’ operative risk. Univariable and multivariable backward stepwise logistic regressions were performed. Overall, 7.6% of patients developed SSIs over three years. The most common SSIs causative microorganism was *Staphylococcus aureus* (7%), whose strains were resistant to penicillin (100%), erythromycin (80%), cotrimoxazole (80%), amikacin (60%) and cefoxitin (60%). Amikacin was the most prescribed antibiotic (36%). Male sex (OR 2.64; 95%CI 1.32–5.30), previous hospitalization (OR 2.15; 95%CI 1.25–3.69), antibiotic prescription during hospitalization before perioperative antibiotic prophylaxis (OR 4.19; 95%CI 2.51–7.00) and postoperative length of stay > 15 days (OR 3.30; 95%CI 1.83–5.95) were identified as significant risk factors. Additionally, preoperative shower significantly increased the SSI risk (OR 4.73; 95%CI 2.72–8.22), which is unconfirmed in the literature so far.

## 1. Introduction

Healthcare-associated infections (HAIs) are infections that are not present or incubating at the time of hospital admission but are acquired during a healthcare facility visit or stay [1]. HAIs contribute to adverse patient outcomes, including prolonged hospitalization, morbidity and mortality, as well as increased treatment costs posing a high financial burden at the individual and health care system level [2,3,4].

Surgical site infections (SSIs) are among the most frequently reported HAIs and may develop post-surgery due to contaminated instruments or environmental conditions of the healthcare facility [5]. In 2017, the European Centre for Disease Prevention and Control (ECDC) reported that SSIs percentage ranged from 0.5% to 10.1% across 13 European countries and varied greatly by the type of surgical procedure and between the countries [5]. In low- and middle-income countries (LMICs), the incidence rate of SSIs was much greater compared to high-income settings [2]. In India, considerable variations in SSIs rates have been reported from different geographical locations within the country and ranged from 1.6% to 38% [6,7,8].

*Staphylococcus aureus* has become the most common cause of SSIs in recent years, as reported by ECDC [5]. *S. aureus* usually originates in patients’ bacterial flora [9] and is responsible for 15–20 % of overall SSIs in hospitals [10,11]. Furthermore, *S. aureus* is the most common causative pathogen of orthopedic implant-associated infections, which can be particularly difficult to treat due to high levels of antibiotic resistance and limited treatment options [12].

Orthopedic surgeries contribute to the overall SSI events in hospitals across the world and remain a challenge for patients and surgeons, requiring the integration of a range of measures before, during and after surgery [13,14,15,16,17]. One of the recommended measures for the prevention of SSIs is a single administration of systemic antibiotics shortly before a surgery, i.e., perioperative antibiotic prophylaxis (PAP) [18,19]. It is estimated that 30–50% of all antimicrobial prescriptions in hospitals are PAP [10].

There are several risk factors associated with SSIs in orthopedic surgery confirmed in the literature, such as male sex, age and length of surgery [20]. According to the US National Healthcare Safety Network (NHSN), the risk index is used to assign surgical patients to one of four categories, from low to high. It is based on the presence of three major risk factors: (1) duration of the operation; (2) wound contamination class; and (3) the American Society of Anesthesiologists (ASA) physical status classification [16,21].

Knowledge of the magnitude of HAIs and SSIs is essential to reducing HAI rates and improving the effectiveness of infection prevention and control measures [22]. Furthermore, in LMICs, the follow-up of surgical patients after discharge is generally neglected and thereby contributes to underestimated SSI rates [4]. Therefore, active and targeted SSI surveillance is recommended [23]. The main goal of surveillance is to provide comprehensive evidence, expert consensus and recommendations, which are to be applied during pre-, intra- and post-operative periods to prevent and reduce the risk of SSIs—one of the objectives of the World Health Organization (WHO) global guidelines for the prevention of SSIs (2016) [24].

In India, there is a lack of a national surveillance system and guidelines on antibiotic use for common infections. Thus, there is a need to conduct recurrent SSIs surveillance at a facility level in order to understand the current situation and develop appropriate recommendations [24]. Active surveillance, audits and feedback have shown an association with the reduction in SSI rates [25]. In order to minimize the incidence rates of orthopedic SSIs, risk factors should be identified at the hospital and community level [26]. The present study aims to assess the incidence and risk factors for SSIs, as well as the profile of common causative SSI pathogens and their antibiotic susceptibility, and to analyze antibiotic use among the operated orthopedic patients in a private, tertiary care hospital in Ujjain city in central India.

## 2. Materials and Methods

### 2.1. Study Setting

The study was conducted at the department of orthopedic surgery in a teaching hospital, located in a rural area of Ujjain city in Madhya Pradesh district in central India. The teaching hospital is a private, tertiary care hospital attached to a medical college run by a non-profit charitable trust. The hospital provides free-of-charge medical services to the community and has a capacity of 750 beds, with 90 beds at the orthopedic department at the time of the study.

### 2.2. Data Collection

Data were collected from August 2013 to April 2016 by trained hospital personnel. Due to the lack of electronic surveillance system in place, data collection was performed using locally developed and validated paper forms. Forms contained information about demographic characteristics of the study population, patient history, provisional and clinical diagnoses, type of performed procedures, surgery outcomes, summary of the laboratory reports of the samples sent for antibiotic susceptibility testing and information of the potential risk factors for SSIs. Information about pre- and post-surgery antibiotic prescriptions was collected using a separate form adopted from previous studies conducted in the same setting [26,27]. Trained data collectors accompanied the orthopedic consultants who clinically identified the SSIs. Respective samples of the identified SSI patients were sent for antibiotic susceptibility testing and laboratory test reports were recorded in the forms. Patients were followed up until discharge with regular updates about antibiotic prescriptions. Data were entered into an Excel file by the trained data entry persons.

### 2.3. Inclusion and Exclusion Criteria

All patients admitted to the orthopedic ward who stayed at least one night were divided into operated and non-operated. In total, 1205 patients were operated and included in the analysis (Figure 1). Out of those, 1004 were operated during the present admission, and 201 were operated during the previous admission, which took place no more than 30 days before the present admission. Of all previously operated patients, 124 were also operated during the present admission tenure. Twenty-nine patients were admitted with a sign of infection and were, therefore, categorized under community-acquired infection and were not included in the SSI group. Operated patients were characterized based on the occurrence of SSIs and antibiotic use (Figure 1).

### 2.4. Data Management and Analysis

SSI occurrence was defined by the Centers for Disease Control and Prevention (CDC) NHSN definition with 30- or 90-day SSI surveillance period, which is determined by the NHSN operative procedure category and the tissue level of an SSI event [21]. SSI surveillance period was one year for patients with implants [28]. SSIs were classified according to NHSN into 3 categories: (i) superficial incisional, (ii) deep incisional, and (iii) organ/ space SSI [21]. Superficial incisional and deep incisional SSIs were further divided into primary and secondary. A primary superficial incision is identified in a patient that has had surgery with one or more incisions. A secondary superficial incision occurs in the secondary incision in a patient that has had an operation with more than one incision. Organ/space SSI is infection that involves any part of the body deeper than the fascial/muscle layers, that is opened or manipulated during the operative procedure [21].The ASA physical status classification system was used to assess the patient’s physiological status to predict the operative risk. According to the ASA classification: ASA I—normal healthy patient (no acute or chronic disease, non-smoker, no or minimal alcohol use); ASA II—patient with mild systemic disease; ASA III—patient with severe systemic disease; ASA IV—patient with a severe systemic disease, which is a constant threat to life.

Standard methods were followed to process the samples sent for culture and susceptibility tests [29]. The inoculated blood agar and McConkey agar plates were incubated at 37 °C for 18–24 hours. Microorganisms were identified by using standard laboratory techniques and the Clinical and Laboratory Standard Institute (CLSI) guidelines [29,30]. The types and number of colony-forming units (CFUs) of identified microorganisms were noted, and percentages of reduction in CFUs and Log10 were calculated for each sample.

The prescribed antibiotics were classified according to the WHO Anatomical Therapeutic Chemical (ATC) classification system [31]. In the study hospital, local prescribing guidelines were not present at the time of the study and consequently, high antibiotic prescribing rates were reported [27].

All information, except prescriptions, was entered into Epidata entry (version 3.1; Epidata software, Odense, Denmark), and the antibiotic prescriptions were entered in Excel. Data were analyzed using Stata 15.1 (Stata Corp., College Station, TX, USA). Continuous variables were presented as median and 25th–75th percentile, and categorical variables were presented as frequencies and percentages. Univariable logistic regression was performed to identify risk factors for SSIs. Statistically significant risk factors (*p*-value < 0.05) were included in multivariable backward stepwise logistic regression analysis. Pearson’s correlation coefficients were calculated for statistically significant risk factors from univariable analysis, and the coefficients that showed high correlation (≥0.5) were excluded from the multivariable analysis in order to avoid multicollinearity and increase the reliability of regression coefficients. Independent variables included in Model 1 were: male sex, ASA II and III scores, previous hospitalization, antibiotic(s) prescribed 14 days before hospital admission, perioperative antibiotic prophylaxis (PAP), antibiotic treatment during hospital stay before PAP, duration of postoperative antibiotic treatment >14 days, postoperative length of stay (LOS) > 15 days, preoperative shower, compound fracture, drain, implant. Akaike information criteria (AIC) and Bayesian information criteria (BIC) were calculated to compare the models and choose the best model.

## 3. Results

Overall, 91/1205 (7.6%) of operated patients developed SSI over three years. SSI incidence rates per year were: 15.5% (August–December 2013), 6.25% (2014), 6.45% (2015), 3.65% (January–April 2016). Significant differences were observed in distribution of potential risk factors between males and females (Appendix A).

Patient-related potential SSI risk factors are presented in Table 1. Median age of all 1205 operated patients was 35 years, and the majority (70%) were male. The physiological status of the majority of SSI patients was calculated as ASA I score (69%), followed by ASA II (24%) and ASA III (7%). A total of 14% of operated patients were previously hospitalized, and 6% were prescribed antibiotic(s) within 14 days before the current admission (Table 1).

Surgery-related potential SSI risk factors are presented in Table 2. The majority of patients who developed SSI (64/91) had closed wounds. For most operated patients (35%), surgery lasted up to one hour. In total, 84% (76/91) of SSI patients had their hair removed by shaving, while 46% (16/91) had a preoperative shower. The median length of preoperative hospital stay did not significantly vary between SSI and non-SSI patients (4 vs. 5 days), whereas the median length of postoperative hospital stay was significantly longer in SSI compared to non-SSI patients (13 vs. 8 days, *p* < 0.001). Drains were used in 41 operated patients, out of which 8 developed SSI. Implants were used in 297 patients, out of which 49 patients developed SSI. A total of 94% operated patients (males—70%; females—30%) were prescribed antibiotics, and all SSI patients were prescribed antibiotics during their hospital stay. PAP was prescribed to 70% of all operated patients and to 46% of patients who developed SSI. Antibiotic(s) before PAP during hospital stay were prescribed to 21% of operated patients (258/1205); out of those, 43 patients developed SSI. A total of 86% of patients were prescribed a postoperative antibiotic, which was given longer than 14 days in 40% (36/91) of SSI patients (Table 2, Appendix A).

Among 91 of operated patients who developed SSIs, 11 (12%) had superficial incisional primary, 2 (2%) superficial incisional secondary, 19 (21%) deep incisional primary, 3 (3%) deep incisional secondary and 53 (58%) organ/space SSI. Table 3 shows that 68 pus or wound samples were sent for culture and susceptibility testing, out of which 15 were culture positive. Two samples showed a mix of two microbial growth of *S. aureus* with *Escherichia coli* and *Klebsiella* spp. The most common microorganism that caused SSIs was *S. aureus* (5/68, 7%), followed by Gram-negative organisms: *Klebsiella* spp. (4/68, 6%), *Pseudomonas* spp. (4/68, 6%) and *E. coli* (2/68, 3%). All strains of *S. aureus* were resistant to penicillin. High resistance was also seen against erythromycin (80%), cotrimoxazole (80%) and amikacin (60%). Three out of five strains of *S. aureus* were resistant to cefoxitin (methicillin-resistant *S. aureus*, MRSA). However, Gram-negative organisms showed more than 50% susceptibility to third-generation cephalosporins (Table 3).

A total of 3030 antibiotic prescriptions were prescribed for 1205 operated orthopedic patients, out of which 11% prescriptions were given to the SSI patients and 89% to the non-SSI patients (Table 4). The most commonly prescribed antibiotic was amikacin (J01GB06, 37%), followed by a combination of ceftriaxone with a β-lactamase inhibitor (J01DD63, 24%) and cefoperazone with a β-lactamase inhibitor (J01DD62, 13%) (Table 4). Additionally, the most prescribed PAP was ceftriaxone or cefoperazone in combination with a β-lactamase inhibitor together with intravenous amikacin.

Table 5 presents the results of the univariable logistic regression analysis, which indicate that the following factors were significantly associated with the risk of developing SSIs: male sex (OR = 3.42, 95% CI = 1.79–6.49), ASA II score (OR = 2.63, 95% CI = 1.57–4.43), previous hospitalization (OR = 4.14, 95% CI = 2.57–6.66), history of antibiotic(s) 14 days before admission (OR = 4.71, 95% CI = 2.59–8.58), PAP (OR = 0.34, 95% CI = 0.21–0.53), antibiotic(s) prescribed during hospital stay before PAP (OR = 3.75, 95% CI = 2.42–5.80), duration of postoperative antibiotic treatment beyond 14 days (OR = 4.23, 95% CI = 2.32–7.69), postoperative LOS beyond 15 days (OR = 5.99, 95% CI = 2.59–13.87), preoperative shower (OR = 3.94, 95% CI = 2.49–6.24), compound fracture (OR = 4.87, 95% CI = 2.21–10.76), the presence of drain (OR = 3.21, 95% CI = 1.43–7.20) and implant (OR = 4.07, 95% CI = 2.64–6.29). Based on these risk factors, three multivariable models were built, out of which Model 3 showed the best combination of AIC and BIC. According to Model 3, the following risk factors were found to be significantly associated with SSIs: male sex (OR 2.64; 95%CI 1.32–5.30), previous hospitalization (OR 2.15; 95%CI 1.25–3.69), antibiotic treatment during hospital stay before PAP (OR 4.19; 95%CI 2.51–7.00), postoperative LOS longer than 15 days (OR 3.30; 95%CI 1.83–5.95), preoperative shower (OR 4.73; 95%CI 2.72–8.22).

## 4. Discussion

In this study, the incidence rate of orthopedic SSIs was 7.6% over three years. Males were 2.64 times more likely to develop SSIs compared to females (95%CI 1.32–5.30). Previously hospitalized patients had 2.15-fold higher odds (95%CI 1.25–3.69) of developing SSIs, whereas patients who received antibiotics during hospital stay before PAP had 4.19-fold higher odds of developing SSIs (95%CI 2.51–7.00). Patients who received antibiotics after surgery longer than 14 days had 4% more chance of developing SSIs (95%CI 1.00–1.09). Patients who stayed in hospital after surgery for longer than 15 days were 3.30 times more likely to develop SSIs (95%CI 1.83–5.95). Patients who showered before the operation had 4.73-fold higher odds of developing SSI (95%CI 2.72–8.22). The most prescribed PAP was third-generation cephalosporin (ceftriaxone—24% or cefoperazone—13%) in combination with β-lactamase inhibitor together with intravenous amikacin (37%). Out of 68 samples sent for culture and susceptibility testing, 22% were culture positive. The most common microorganism that caused SSIs was *S. aureus* (7%), and 60% of its strains were resistant to cefoxitin (MRSA).

SSI incidence of 7.6% over 3 years is in the range of overall SSI incidences reported in the EU countries (0.5–10.1%) [5]. However, in India, reported SSI rates largely vary from 1.6% to 38% [8]. A study from Madhya Pradesh in 2014 reported a lower SSI rate (2.1%) in orthopedic patients compared to our study [10]. In general, studies show that orthopedic procedures have somewhat lower SSI rates compared to other procedures in both high- and middle-income countries, as reported by studies in New Zealand (1.3%), China (2.18%) and Jordan (2.8%) [13,32,33]. A systematic review from 57 hospitals across the world reported orthopedic SSI rate of 2.7% [11]. The difference in the incidence rates can partially be attributed to higher standards and stricter policies for delivering care in high- and some middle-income countries.

In our study, *S. aureus* was the most common pathogen causing SSIs, responsible for 33% of the culture-positive samples. Likewise, studies from New Zealand [32] and India [10] reported *S. aureus* to be the main causative organism of orthopedic SSIs, responsible for 54% and 29% of the culture-positive samples, respectively. However, in a study from China, *Coagulase-negative Staphylococcus* was the predominant SSIs causative pathogen (42.8%) in orthopedic surgery, followed by *S. aureus* (11.4%) [13]. Moreover, in our study, 60% of *S. aureus* samples were MRSA. More than 50 % of *S. aureus* HAIs in Europe and the US are caused by MRSA, which is becoming increasingly challenging to treat due to antibiotic resistance and limited treatment options [11].

In orthopedic surgery, PAP is considered to be one of the most effective measures to reduce the risk of SSIs [34]. In the US and New Zealand guidelines the most widely recommended PAP for orthopedic procedures is cefazolin [32,35]. In our study, the most used PAP was third-generation cephalosporin (ceftriaxone or cefoperazone in combination with beta-lactamase inhibitor) together with intravenous amikacin. Different choices of PAP might be explained with different prevalent bacteria, susceptibility patterns and operating theatre conditions in Indian setting [34]. No orthopedic prescribing guidelines were in place in the teaching hospital at the time of the study. Given that 20% and 47% of our culture-positive bacterial isolates were resistant to ceftriaxone and amikacin, respectively, appropriate modifications to the usual choice of PAP are suggested to prevent SSIs more efficiently.

In our study, male sex was shown to be significantly associated with SSIs. This is in line with previous research which demonstrated that men, in general, are more likely to develop SSIs than women [36,37]. A German study suggested that male patients undergoing orthopedic and trauma surgeries had significantly higher SSI incidence rates than female patients [38]. This might be explained by men having higher colonization rates of *S. aureus,* the most prevalent SSI-causative bacteria [38].

Postoperative LOS longer than 15 days and previous hospitalization significantly increased the risk of SSIs. Previous surgery was confirmed as a risk factor by previous research [13], especially in the case of spinal surgery [39]. Postoperative LOS was also identified as a risk factor for orthopedic SSIs by a cohort study from Jordan [33]. Previous hospitalization might also be associated with increased LOS [40]. In our study, the median LOS was significantly higher in SSI patients (13 days) compared to non-SSI patients (8 days). A Swedish study showed that 42% of all adverse events in orthopedic surgery prolong the LOS for an average of 6.1 days [41]. One study from India showed that the maximum median LOS was in surgical oncology patients (31.5 days), followed by orthopedic surgery patients (14 days) [42]. 

Antibiotic treatment during hospital stay before PAP was significantly associated with the risk of developing SSIs. The patients who needed prolonged preoperative and postoperative antibiotic treatment were mostly the patients with implants or osteomyelitis who had come to the hospital with signs of delayed or late infection (e.g., pus, swelling or abscess) [43]. Prolonged antibiotic treatment contributes to the development of antibiotic resistance [44], which has most likely contributed to the development of SSIs [45].

Preoperative shower was found to significantly increase the risk of orthopedic SSIs. The literature on the benefit of antiseptic preoperative shower is controversial. Some studies list preoperative shower as a protective factor that reduces the incidence of SSIs, which is explained by the reduction in microbial colonization of skin [46,47]. On the other hand, certain studies found no clinically relevant benefit of preoperative chlorhexidine showers [47,48]. Contrary to these findings, the results of our study suggest that preoperative shower is a significant risk factor for SSIs. This might be due to the fact that in the teaching hospital, patients were only advised to take shower or bath before surgery, hence we do not know if patients had actually taken a shower and with what (just water, soap, chlorhexidine). Additionally, the majority of patients (86%) had a closed fracture; therefore, they might not have showered the broken limb properly or at all. Furthermore, the microbiological quality of water that people use for washing in the Ujjain district has been questioned earlier; therefore, a similar study is proposed to check the water quality in the setting [49]. 

Most SSIs (60%) occur after hospital discharge [50]. The time until SSI onset tends to be among the longest in orthopedic surgeries because of the risk of postponed infection associated with the implants [11]. A systematic review showed that SSIs occur, on average, 33.5 days after orthopedic surgery [11]. In our study, the follow-up was performed either 30 or 90 days after surgery, or after one year for patients with implants. Another study showed that most of operated patients (>75%) did not return to hospital for follow-up after surgery, and calling the unreturned patients was the only choice left [51]. However, follow-up was difficult when patients did not have a direct number to contact and it was not always feasible to send a text message, as most of the patients were from rural India and not educated enough to read the text messages [51]. In our study, special care was taken to avoid lost to follow-up, by noting the residential address and two separate telephone/mobile numbers in the form. Patients who failed to visit the hospital after one month of surgery were followed up by phone. As per hospital policy, free medicines, x-rays and laboratory investigations might have acted as incentives and attracted the patients to come for follow-ups. Despite all the efforts, a chance of underestimation of SSIs cannot be denied, as the postoperative follow-up was only performed in 27% of all the patients. Nevertheless, the presumption of orthopedic staff was that if patients had an infection or postoperative complications, they would have most likely come to the hospital for a follow-up. 

This study had a long follow-up time, sufficient to identify patients who developed SSIs, including those with late implant infection. However, a relatively high loss to follow-up might have led to an underestimation of the SSI rate. Additionally, there is a lack of information about how preoperative shower was carried out. A relatively small sample size might have affected the multivariable analysis of potential confounders and risk factors for SSIs. However, based on the formula by Pourhoseingholi et al. [52], the minimum sample size required for the small expected prevalence of outcome <10% (7.55% in this study), the precision of 0.017 and 0.05 alpha level of significance is 928 participants; therefore, the sample size in our study exceeded the minimum number of required participants. 

Based on the results of this study, appropriate modification of the current choice of PAP is advised to reduce the incidence of SSIs. Furthermore, a community-based study is recommended to complement this hospital-based study in order to identify more accurately the SSI incidence rate. Additionally, further research is needed to investigate the ways of performing preoperative shower and the reasons behind it being a risk factor, and to check the water quality in the setting.

## 5. Conclusions

The SSI incidence rate of 7.6% over three years in the present study was relatively low compared to the reported incidence range for India, yet higher than the reported SSI incidences for orthopedic surgical procedures in high- and middle-income countries. The most common SSI-causative pathogen was *S. aureus* and the most prescribed PAP was third-generation cephalosporin in combination with intravenous amikacin. Factors that significantly increased the risk of orthopedic SSIs were male sex, previous hospitalization, antibiotic treatment during hospital stay before PAP and postoperative LOS longer than 15 days. Preoperative shower was also found to be a significant risk factor for SSIs, which is undocumented in the literature so far, to the best of our knowledge. Further studies are needed to confirm this finding and explore the possible explanations behind it. The identification of SSI incidences and risk factors in orthopedic surgery wards supports overall measures to prevent and mitigate SSIs in hospitals.

## Figures and Tables

**Figure 1 antibiotics-11-00748-f001:**
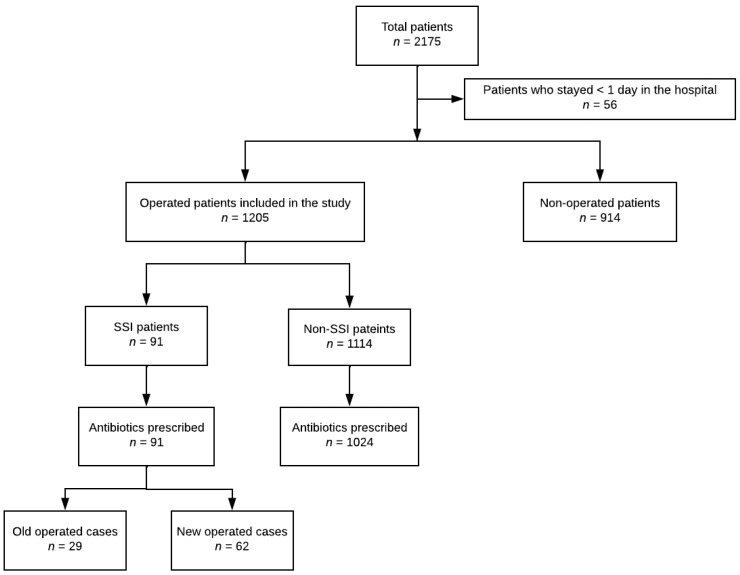
Flowchart of the study population selection. SSI: surgical site infection.

**Table 1 antibiotics-11-00748-t001:** Patient-related potential risk factors for orthopedic surgical site infections in the teaching hospital, Ujjain, central India.

	All Operated Patients *n* = 1205 (%)	SSI Patients *n* = 91 (%)	Non-SSI Patients *n* = 1114 (%)
**Age, median (25–75th), years**	35 (19–50)	35 (22–50)	35 (18–50)
**Age, years**			
≤18	301 (25)	18 (20)	283 (25)
19–60	760 (63)	64 (70)	696 (62)
>60	144 (12)	9 (10)	135 (12)
**ASA score**			
**ASA I**	1013 (84)	63 (69)	950 (85)
**ASA II**	148 (12)	22 (24)	126 (11)
**ASA III**	43 (4)	6 (7)	37 (3)
**ASA IV**	1 (0)	0	1 (0)
**Antibiotic prescribed 14 days before hospital admission**	73 (6)	17 (19)	56 (5)
**Previous hospitalization**	173 (14)	33 (36)	140 (13)

SSI = surgical site infection, ASA = American Society for Anesthesiologists.

**Table 2 antibiotics-11-00748-t002:** Surgery-related potential risk factors for orthopedic surgical site infections in the teaching hospital, Ujjain, central India.

	All Operated Patients *n* = 1205 (%)	SSI Patients *n* = 91 (%)	Non-SSI Patients *n* = 1114 (%)
**Type of wound ^a^**			
Closed	1034 (86)	64 (70)	970 (87)
Compound fracture	37 (3)	9 (10)	28 (3)
Clean	3 (0)	0	3 (0)
Contaminated	23 (2)	3 (3)	20 (2)
**Nature of surgery ^a^**			
Elective	1113 (92)	80 (88)	1033 (93)
Emergency	17 (1)	2 (2)	15 (1)
**Duration of surgery ^a^, min**			
≤60	425 (35)	40 (44)	385 (35)
61–120	375 (31)	22 (24)	353 (32)
>120	208 (17)	13 (14)	195 (18)
**Hair removal method ^a^**			
Shaving	1057 (88)	76 (84)	981 (88)
Clipping	2 (0)	0	2 (0)
**Preoperative shower**	267 (22)	42 (46)	225 (20)
**Preoperative LOS, median (25–75th), days**	5 (3–9)	4 (2–8)	5 (3–9)
**Preoperative LOS ^a^, days**			
1–3	325 (27)	24 (26)	301 (27)
4–7	379 (31)	28 (31)	351 (32)
8–15	312 (26)	16 (18)	296 (27)
>15	90 (7)	9 (10)	81 (7)
**Postoperative LOS, median (25–75th), days**	8 (3–14)	13 (4–21)	8 (3–14)
**Postoperative LOS ^a^, days**			
1–3	223 (19)	7 (8)	216 (19)
4–7	239 (20)	8 (9)	231 (21)
8–15	440 (37)	28 (31)	412 (37)
>15	203 (17)	33 (36)	170 (15)
**Oxygen support**	1031 (86)	73 (80)	958 (86)
**Blood transfusion**	405 (34)	27 (30)	378 (34)
**Drain**	41 (3)	8 (9)	33 (3)
**Implant**	297 (25)	49 (54)	248 (22)
**Antibiotic prescription**	1133 (94)	91 (100)	1042 (94)
**PAP**	840 (70)	42 (46)	798 (72)
**Antibiotic during hospital stay before PAP**	258 (21)	43 (47)	215 (19)
**Duration of antibiotic treatment before PAP, days**			
1–7	186 (15)	29 (32)	157 (14)
8–14	44 (4)	8 (9)	36 (3)
>14	28 (2)	6 (7)	22 (2)
**Postoperative antibiotic**	1036 (86)	75 (82)	961 (86)
**Duration of postoperative antibiotic, days**			
1–7	440 (37)	17 (19)	423 (38)
8–14	374 (31)	24 (26)	350 (31)
>14	248 (21)	36 (40)	212 (19)
**Antibiotic duration, median (25–75th), days**	12 (4–16)	24 (8–36)	11 (4–15)
**Total antibiotic duration ^a^, days**			
1–7	384 (32)	21 (23)	363 (33)
8–14	319 (26)	19 (21)	300 (27)
>14	391 (32)	50 (55)	341 (31)

^a^ For the variables where the number of patients does not correspond to the total number of patients in the group, that information for the rest of the patients is missing in the data record. SSI = surgical site infection, PAP = perioperative antibiotic prophylaxis, LOS = length of stay.

**Table 3 antibiotics-11-00748-t003:** Antibiotic susceptibility patterns of the bacterial isolates in orthopedic surgical site infections in the teaching hospital, Ujjain, central India.

Antibiotics Tested	Gram-Positive Organisms	Gram-Negative Organisms			
	*S. aureus* (*n* = 5)	*Pseudomonas* (*n* = 4)	*Klebsiella* (*n* = 4)	*E. coli* (*n* = 2)	Total
Penicillin	5	-	-	-	-
Erythromycin	4	-	-	-	-
Ciprofloxacin	3	3	1	1	5/10
Cefoxitin	3	-	1	1	2/6
Tetracycline	2	-	3	1	4/6
Cotrimoxazole	4	-	2	2	4/6
Vancomycin	-	-	-	-	-
Linezolid	-	-	-	-	-
Clindamycin	-	-	-	-	-
Amikacin	3	3	1	0	4/10
Gentamycin	3	3	1	1	5/10
Ampicillin	-	-	3	1	4/6
Amoxiclav	-	-	2	1	3/6
Piperacillin Tazobactam	-	3	1	0	4/10
Cefuroxime	-	-	2	1	4/6
Cefepime	-	3	2	1	6/10
Cefotaxime	-	-	2	1	3/6
Ceftriaxone	-	-	2	1	3/6
Ceftazidime	-	3	2	1	6/10
Meropenem	-	1	0	0	1/10
Aztreonam	-	3	0	1	4/10

Susceptibility to Colistin in Gram-negative organisms was 100%.; one *Klebsiella* isolate was the extended-spectrum β-lactamase producer.

**Table 4 antibiotics-11-00748-t004:** Antibiotic prescriptions during hospital stay at the orthopedic ward in the teaching hospital, Ujjain, central India.

Antibiotics Groups/Subgroups/Substances with ATC Codes	Total Prescriptions *n* = 3030 (%)	Prescriptions for SSI Patients *n* = 332 (%)	Prescriptions for Non-SSI Patients *n* = 2698 (%)
**Tetracyclines; J01A**			
Tetracyclines; J01AA02	5 (0)		5 (0)
**β-lactams, Penicillins; J01C**			
Combinations of penicillins, incl. β-lactam inhibitors; J01CR02	281 (9)	36 (11)	245 (9)
J01CR05	3 (0)	2 (1)	1 (0)
J01CR50	1 (0)	1 (0)	
**Other β-lactams; J01D**			
Second-generation cephalosporins; J01DC02	3 (0)	1 (0)	2 (0)
J01DC10	1 (0)		1 (0)
Third- generation cephalosporins; J01DD01	43 (1)	9 (3)	34 (1)
J01DD04	12 (0)		12 (0)
J01DD08	1 (0)		1 (0)
J01DD12	2 (0)		2 (0)
J01DD13	8 (0)		8 (0)
J01DD62	380 (13)	27 (8)	353 (13)
J01DD63	738 (24)	70 (21)	668 (25)
Carbapenems; J01DH51	1 (0)		1 (0)
**Sulfonamides and Trimethoprim; J01E**			
Combinations of sulfonamides and trimethoprim; J01EE01	3 (0)		3 (0)
**Macrolides, Lincosamides and Streptogramins; J01F**			
Lincosamides; J01FF01	6 (0)	3 (1)	3 (0)
**Aminoglycosides; J01G**			
Other aminogylcosides; J01GB03	23 (1)	8 (2)	15 (1)
J01GB06	1107 (37)	96 (29)	1011 (38)
**Quinolones; J01M**			
Fluoroquinolones; J01MA01	1 (0)		1 (0)
J01MA02	79 (3)	19 (6)	60 (2)
J01MA06	3 (0)	1 (0)	2 (0)
**Combinations of antibacterials; J01R**			
Combinations of antibacterials; J01RA75	1 (0)		1 (0)
**Other antibacterials; J01X**			
Imidazole derivatives; J01XD01	84 (3)	22 (7)	62 (2)
Other antibacterials; J01XX08	244 (8)	37 (11)	207 (8)

SSI = surgical site infection, ATC = Anatomical Therapeutic Chemical classification.

**Table 5 antibiotics-11-00748-t005:** Univariable and multivariable analyses of risk factors associated with orthopedic surgical site infections.

Risk Factor		Univariable Analysis			Multivariable Analysis								
					Model 1			Model 2			Model 3		
					AIC = 454, BIC = 523			AIC = 482, BIC = 512			AIC = 447, BIC = 487		
		OR	95% CI	*p*-Value	OR	95% CI	*p*-Value	OR	95% CI	*p*-Value	OR	95% CI	*p*-Value
Sex	Female	1											
	Male	3.42	1.79–6.49	0.000	2.57	1.25–5.29	0.010	2.93	1.48–5.77	0.002	2.64	1.32–5.30	0.006
Age, years	≤18	1.00											
	19–60	1.45	0.84–2.48	0.182									
	>60	1.05	0.46–2.39	0.911									
ASA score	ASA I	1											
	ASA II	2.63	1.57–4.43	0.000	1.30	0.67–2.49	0.437						
	ASA III	2.45	0.99–6.01	0.051	2.08	0.76–5.72	0.156						
Previous hospitalization		4.14	2.57–6.66	0.000	1.65	0.85–3.19	0.139				2.15	1.25–3.69	0.006
Antibiotic prescribed 14 days before hospital admission		4.71	2.59–8.58	0.000	1.45	0.61–3.42	0.400						
PAP		0.34	0.21–0.53	0.000	1.11	0.52–2.34	0.789						
Antibiotic treatment during hospital stay before PAP		3.75	2.42–5.80	0.000	3.93	2.33–6.63	0.000	3.92	2.40–6.43	0.000	4.19	2.51–7.00	0.000
Duration of preoperative antibiotic, days	1–7	1											
	8–14	1.2	0.51–2.85	0.674									
	>14	1.48	0.55–3.96	0.438									
Postoperative antibiotic		0.75	0.42–1.31	0.311									
Duration of postoperative antibiotic, days	1–7	1											
	8–14	1.71	0.90–3.23	0.100									
	>14	4.23	2.32–7.69	0.000	1.05	1.00–1.09	0.043	1.05	1.01–1.09	0.028	1.04	1.00–1.09	0.051
Preoperative LOS, days	1–3	1											
	4–7	1.00	0.57–1.76	0.999									
	8–15	0.68	0.35–1.30	0.243									
	>15	1.39	0.62–3.12	0.419									
Postoperative LOS, days	1–3	1											
	4–7	1.07	0.38–2.99	0.900									
	8–15	2.10	0.90–4.88	0.086									
	>15	5.99	2.59–13.87	0.000	3.03	1.65–5.58	0.000	2.95	1.67–5.20	0.000	3.30	1.83–5.95	0.000
Preoperative shower		3.94	2.49–6.24	0.000	4.14	1.99–8.56	0.000	5.49	3.29–9.16	0.000	4.73	2.72–8.22	0.000
Hair removal	Not done	1.00											
	Previous night	0.65	0.36–1.19	0.161									
	Same day	0.56	0.15–2.03	0.375									
	Shaving	0.59	0.33–1.08	0.087									
Type of fracture	Closed	1											
	Compound	4.87	2.21–10.76	0.000	1.97	0.73–5.35	0.182						
Nature of surgery	Elective	1											
	Emergency	1.72	0.39–7.66	0.476									
Duration of surgery, min	≤60	1.00											
	61–120	0.60	0.35–1.03	0.064									
	>120	0.64	0.34–1.23	0.180									
Blood transfusion		0.88	0.54–1.43	0.601									
Oxygen support		0.75	0.29–1.93	0.547									
Drain		3.21	1.43–7.20	0.005	1.83	0.74–4.50	0.189				1.73	0.71–4.22	0.231
Implants		4.07	2.64–6.29	0.000	1.34	0.71–2.50	0.366						

PAP = perioperative antibiotic prophylaxis, LOS = length of stay.

## Data Availability

As per the policy of the institute, the metadata of any research are not shared with the general public. This is to protect the patients’ confidentiality and hospital and hospital staff safety concerning the medical, ethical and legal issues. However, the data can be made available to the researchers who meet the criteria to access the confidential data via the Chairman of the Ethics Committee, R.D. Gardi Medical College, Agar Road, Ujjain, Madhya Pradesh, India, 456006 (email: iecrdgmc@yahoo.in), by giving all details of the article. A request can be made by quoting the ethical approval number: 311/2013.

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
