# Peer review of "Antibiotic Use, Incidence and Risk Factors for Orthopedic Surgical Site Infections in a Teaching Hospital in Madhya Pradesh, India"

_antibiotics, 2022, doi:10.3390/antibiotics11060748_

Round 1

Reviewer 1 Report

The article is well designed and written. The data analyses are thorough. However, the data representation or the table organization needs to be improved. The tables are a bit difficult to read. It would be great if the authors could re-organize the data in table to make it easier to read.

Author Response

Response to Reviewer 1 Comments

Point 1: The article is well designed and written. The data analyses are thorough. However, the data representation or the table organization needs to be improved. The tables are a bit difficult to read. It would be great if the authors could re-organize the data in table to make it easier to read.

Response 1: Thank you for this comment. Tables 1 and 2 have been simplified now- the distribution of potential risk fators by sex has been removed. Original Tables 1 and 2 with the distribution of potential risk factors by sex have been put in the Appendix.

Reviewer 2 Report

Congratulations on your data. The layout of manuscript and scientific soundness is satisfied.

Please append your IRB approval.

As you mentioned that your result may be underestimated in the conclusion, "The SSI incidence rate of 7.6% over three years in the present study was relatively low 391 compared to the reported incidence range for India, yet higher than reported SSI incidences for orthopaedic surgical procedures in high- and middle-income countries."

I would suggest you to make a figure to delineate the trend of SSI incidence per year in your cohort (2013-2016). What's the impact of this underestimation in your study should be emphasized appropriately.

Author Response

Response 1: Thank you for this input. IRB approval has now been attached. This study was a part of larger Indo-Swedish collaborative research project under acronym APRIAM.

Response 2: Thank you for this suggestion. I have enclosed a figure and table with calculated SSI incidence rates per year. Please note that for 2013 and 2016, we did not have complete data of the entire years. Table has been attached in the Appendix, and this sentence was added to the Results section: “SSI incidence rates per year were: 15.5% (Aug-Dec 2013), 6.25% (2014), 6.45% (2015), 3.65% (Jan- Apr 2016).”

Reviewer 3 Report

Thank you for the opportunity to review this study. 

The study paints a good picture of the problem of orthopaedic surgical site infections (SSIs) in a hospital in a rural area of India. I think this description is very useful to understand the management differences and limitations that may characterise certain demographic areas.

The study is in my opinion perfectly conducted. The introduction is informative. The methods are extremely well explained. The statistics are methodologically correct. The results are clear. The discussion is detailed and the limitations of the study are adequately described. The conclusions are consistent with the results. I congratulate the authors on the honesty of this work.

I think the study is worthy of publication. I only have a couple of suggestions that I think can improve the usability of the text:

1) on line 44 you said: 'Surgical site infections (SSIs) are the most frequently reported HAIs'. Please insert a reference for this statement.

2) I think too much space has been given graphically to the analysis of the different incidence of SSIs according to gender (male/female). I would consider simplifying Tables 1 and 2 a lot, providing most of the data in those tables as supplementary material.

Thank you.

Author Response

Response to Reviewer 3 Comments

Point 1: Thank you for the opportunity to review this study. The study paints a good picture of the problem of orthopaedic surgical site infections (SSIs) in a hospital in a rural area of India. I think this description is very useful to understand the management differences and limitations that may characterise certain demographic areas.

The study is in my opinion perfectly conducted. The introduction is informative. The methods are extremely well explained. The statistics are methodologically correct. The results are clear. The discussion is detailed and the limitations of the study are adequately described. The conclusions are consistent with the results. I congratulate the authors on the honesty of this work.

I think the study is worthy of publication. I only have a couple of suggestions that I think can improve the usability of the text:

1) on line 44 you said: 'Surgical site infections (SSIs) are the most frequently reported HAIs'. Please insert a reference for this statement.

Response 1:

Thank you for this input. The references have been added now in the text as follows: “Surgical site infections (SSIs) are among the most frequently reported HAIs and may develop post-surgery due to contaminated instruments or environmental conditions of the healthcare facility [5]. In 2017, the European Centre for Disease Prevention and Control (ECDC) reported that the SSIs percentage ranged from 0.5%-10.1% across 13 European countries and varied greatly by the type of the surgical procedure and between the countries [5].”

[5] European Centre for Disease Prevention and Control. Healthcare-associated infections: surgical site infections. Annual epi-demiological report for 2017 [Internet]. Stockholm: ECDC; 2019 [cited 2021 May 21]. Available from: https://www.ecdc.europa.eu/sites/default/files/documents/AER_for_2017-SSI.pdf

Point 2:  I think too much space has been given graphically to the analysis of the different incidence of SSIs according to gender (male/female). I would consider simplifying Tables 1 and 2 a lot, providing most of the data in those tables as supplementary material. Thank you.

Response 2: Thank you for this suggestion. Tables 1 and 2 have been simplified now- the distribution of potential risk fators by sex has been removed. Original Tables 1 and 2 with the distribution of potential risk factors by sex have been put in Appendix.